Text mining for identification of biological entities related to antibiotic resistant organisms

Fortunato Costa Kelle 1
Almeida Araújo Fabrício 2 3
http://orcid.org/0000-0002-8566-3238 Morais Jefferson 4
Lisboa Frances Carlos Renato 1
http://orcid.org/0000-0002-8032-1474 Ramos Rommel T. J. 5 rommelramos@ufpa.br
1 Programa de pós-graduação em Engenharia Elétrica, Universidade Federal do Pará , Belém, Pará , Brazil
2 Biological Science Institute, Universidade Federal do Pará , Belém, Pará , Brazil
3 Universidade Federal Rural da Amazônia , Belém, Pará , Brazil
4 Universidade Federal do Pará , Belém, Pará , Brazil
5 Biological Science Institute, Universidade Federal do Para , Belém, Pará , Brazil
Piccolo Stephen
Electronic publication date: 2022 May 5
Publication date: 2022
Volume: 10
Electronic Location ID: e13351
Received 2021 Jul 13; Accepted 2022 Apr 7
Copyright: © 2022 Fortunato Costa et al.
Copyright year: 2022
Copyright holder: Fortunato Costa et al.
License: This is an open access article distributed under the terms of the Creative Commons Attribution License, which permits unrestricted use, distribution, reproduction and adaptation in any medium and for any purpose provided that it is properly attributed. For attribution, the original author(s), title, publication source (PeerJ) and either DOI or URL of the article must be cited.
License URL: https://creativecommons.org/licenses/by/4.0/

Keywords: Antimicrobial resistance, Biological literature, Text mining, Machine learning

Funding: Dean of Research and Graduate Studies (PROPESP) 06/2021 This work was supported by the Dean of Research and Graduate Studies (PROPESP) through the Qualified Publication Support Program (PAPQ) (notice 06/2021). The funders had no role in study design, data collection and analysis, decision to publish, or preparation of the manuscript.

==============================
Antimicrobial resistance is a significant public health problem worldwide. In recent years, the scientific community has been intensifying efforts to combat this problem; many experiments have been developed, and many articles are published in this area. However, the growing volume of biological literature increases the difficulty of the biocuration process due to the cost and time required. Modern text mining tools with the adoption of artificial intelligence technology are helpful to assist in the evolution of research. In this article, we propose a text mining model capable of identifying and ranking prioritizing scientific articles in the context of antimicrobial resistance. We retrieved scientific articles from the PubMed database, adopted machine learning techniques to generate the vector representation of the retrieved scientific articles, and identified their similarity with the context. As a result of this process, we obtained a dataset labeled “Relevant” and “Irrelevant” and used this dataset to implement one supervised learning algorithm to classify new records. The model’s overall performance reached 90% accuracy and the f-measure (harmonic mean between the metrics) reached 82% accuracy for positive class and 93% for negative class, showing quality in the identification of scientific articles relevant to the context. The dataset, scripts and models are available at https://github.com/engbiopct/TextMiningAMR.

Introduction

Antibiotics are the most successful drugs of the last 100 years, responsible for saving countless lives and enabling modern medical procedures that would otherwise be unthinkable. However, all antibiotics derived from secondary or fully synthetic microbial metabolism products are subject to resistance (Wright, 2011).

Antimicrobial resistance (AMR) has been increasingly recognized as an important public health problem worldwide, considering that infections caused by multidrug-resistant organisms (MDR) result in a significant increase in mortality and cause a tremendous economic burden (Tran, Munita & Arias, 2015). In addition to the costs of rising hospital admission rates, it is estimated that by 2,050 there will be an economic loss of $100 trillion in global antibiotic production (Review on Antimicrobial Resistance, 2016).

In recent years, the scientific community has intensified efforts to combat this problem by making available a wide range of public databases specific to AMR, such as: National Database of Antibiotic Resistant Organisms (NDARO) (Annual Reports for NLM Program and Services, 2016), Comprehensive Antibiotic Resistance Database (CARD) (Alcock et al., 2020), Resfinder (Zankari et al., 2012), ResfinderFG (Munk et al., 2018), Resfams (Gibson, Forsberg & Dantas, 2015), Antibiotic Resistance Genes Database (ARDB) (Liu & Pop, 2009), MEGARes (Lakin et al., 2017), Antibiotic Resistance Gene Annotation (ARG-ANNOT) (Gupta et al., 2014), Mustard (Ruppe et al., 2019), Functional Antibiotic Resistance Metagenomic Element (FARME database) (Wallace et al., 2017), SARG (v2) (Yin et al., 2018), Lahey list of β-lactamases (Bush & Jacoby, 2010), β-Lactamase Database (BLDB) (Naas et al., 2017), Lactamase Engineering Database (LacED) (Thai, Bos & Pleiss, 2009; Thai & Pleiss, 2010), Comprehensive β-Lactamase Molecular Annotation Resource (CBMAR) (Srivastava et al., 2014), among others, which are frequently used as reference databases, with gene sequences related to antimicrobial resistance and metadata that enrich the characterization of sequences. Due to the increase in the volume of literature related to biological and health sciences, the curation process has become challenging for researchers and biocurators who use these databases as a source of research, mainly due to the time required to locate relevant information about biological entities related to antibiotic-resistant organisms. Even queries in specialized databases in biomedical literature such as PubMed (scientific and medical abstracts/citations), PubMed Central (full-text journal articles), NLM Catalog (index of NLM collections), Books (books and reports), and MeSH (ontology used for PubMed indexing) (Sayers et al., 2019), tends to make document selection difficult due to a large amount of retrieved items.

In this context, the adoption of text mining (TM) techniques are viable alternatives (Wei, Kao & Lu, 2013), as they can help in different stages of the standard biocuration workflow. According to (Hirschman et al., 2012), the steps are: Selection: search for articles relevant to the curation.

Identification and standardization of bioentities: detection of mentions of bioentities relevant to the curatorship; for example, genes, proteins, or small molecules, linked to unique identifiers from databases such as UniProt, EntrezGene, or ChEBI.

Detection of annotation events: identification and encoding of events, such as descriptions of protein-protein interactions, characterizations of gene products in terms of cellular location, molecular function, involvement in the biological process and phenotypic effect.

Evidence qualifier association: association of experimental evidence that supports the annotation event performed due to biocuration efforts.

Completion and verification of the database record.

TM technologies combine knowledge resources such as controlled vocabularies, taxonomies, and ontologies with linguistic analysis and machine learning to deal with language variations and extract not only terms from the text but also relationships between terms (Chaix et al., 2018).

In the last decade, several applications in the biomedical area were developed (Fleuren et al., 2011; Fontelo, Liu & Ackerman, 2005; Perez-Iratxeta, Bork & Andrade, 2001; Lewis et al., 2006; Fontaine et al., 2009; States et al., 2009; Huang et al., 2013; Hokamp & Wolfe, 2004; Plikus, Zhang & Chuong, 2006; Becker et al., 2003; Douglas, Montelione & Gerstein, 2005; Brancotte et al., 2011; De et al., 2010; Smalheiser, Zhou & Torvik, 2008; Chen & Sharp, 2004; Li et al., 2013; Glynn, Kerin & Sweeney, 2010; Xuan et al., 2007; Giglia, 2011; Tsuruoka et al., 2011; Fernandez, Hoffmann & Valencia, 2007; Raja, Subramani & Natarajan, 2013; Pafilis et al., 2009; Rebholz-Schuhmann et al., 2008; Plake et al., 2006; Soldatos et al., 2010; Franceschini et al., 2013) using one or more of the following TM steps: (i) retrieving textual resources relevant to a particular subject of interest, a process known as information retrieval (IR), (ii) detect the occurrence of specific keywords of interest and the relationships between these keywords and (iii) infer new relationships based on known facts, and this step is called knowledge discovery (KD) (Fleuren & Alkema, 2015).

One of the most used machine learning techniques in knowledge discovery, especially in document screening, is text classification. However, supervised classification requires the prior labeling of a training set, a non-trivial task for human curators, as it requires a lot of time and effort.

In this sense, Suomela & Andrade (2005) propose a methodology for automatic (binary) classification of large volumes of data, adopting the word counting technique, known as the bag of words (BOW) (Manning, Raghavan & Schütze, 2008), a textual representation that composes the vector space model (Salton, Wong & Yang, 1975), where documents are converted into vectors of words. A weighting scheme is applied to each word, which can be a simple word count or a metric such as Term Frequency-Inverse Document Frequency (TF-IDF) (Manning, Raghavan & Schütze, 2008; Paik, 2013; Chen, 2017) and based on the arithmetic mean of the weights of these words, text summaries are classified as relevant or irrelevant.

This methodology was implemented from specific abstracts of an area of interest, extracted from Pubmed, and inspired the development of the MedlineRanker application (Fontaine et al., 2009) and served as a baseline for this study, which instead of the bag of words, adopts a representation approach based on neural networks (Bengio et al., 2006), called Paragraph Vector-Distributed Memory (PV-DM) (Le & Mikolov, 2014), capable of revealing semantic characteristics between documents, a property that makes this approach useful for many natural language processing (NLP) tasks and justifies its wide use in works involving natural language understanding (Collobert & Weston, 2008; Zhila et al., 2013), machine translation (Mikolov, Le & Sutskever, 2013; Zou et al., 2013), image comprehension (Frome et al., 2013) and relational extraction (Socher et al., 2013a).

The study of antimicrobial resistance genes is essential to public health. However, there is challenging to handle and extract the amount of available data of scientific and medical manuscripts from public databases without computational methods. Thus, this work proposes an unsupervised learning-based TM approach for ranking the relevance of articles on AMR context to generate a set of training, accurate enough to generalize new data, maximizing the efficiency of the supervised classifiers.

Materials and Methods

Labeling pipeline

Figure 1 shows the TM steps implemented in this work in order to label the data.

Figure 1 Proposed TM model.

Steps (A) and (B) include retrieving the information. Steps (C) and (D) include the recognition of entities and the discovery of knowledge, resulting in a metric (cosine similarity) responsible for determining the binary classification performed in step (E).

Information retrieval

An Application Programming Interface (API) was implemented to retrieve a collection of relevant articles in the Drug Resistance, and Microbial domain through the Pubmed Central (PMC) database, which contains free full-text files of the Library’s of Medicine and the US National Institutes of Health (NIH/NLM) biomedical literature are available. In the API, the E-Search and E-Fetch tools from the E-utilities package were used, which provide a structured interface for accessing the Entrez system, the NCBI database system, which currently includes 38 databases, covering a variety of biomedical data, including nucleotide and protein sequences, gene records, three-dimensional molecular structures, and biomedical literature (NCIBI Homepage, https://www.ncbi.nlm.nih.gov/books/NBK3827/). Table 1 presents the set of parameters incorporated into the search (E-Search).

Table 1 Parameters E-Search PubMed central.

Parameters	Value	
URL	https://eutils.ncbi.nlm.nih.gov/entrez/eutils/esearch.fcgi?db=pmc&term=%22drug%20resistance,%20microbial%22[MeSH%20Terms]%20OR%20(%22drug%22[All%20Fields]%20AND%20%22resistance%22[All%20Fields]%20AND%20%22microbial%22[All%20Fields])%20OR%20%22microbial%20drug%20resistance%22[All%20Fields]%20OR%20(%22drug%22[All%20Fields]%20AND%20%22resistance%22[All%20Fields]%20AND%20%22microbial%22[All%20Fields])%20OR%20%22drug%20resistance,%20microbial%22[All%20Fields])%20AND%20%22open%20access%22[filter]&retmax=10000	
db	PMC (full text articles)	
Term	(“drug resistance, microbial”[MeSH Terms] OR (“drug”[All Fields] AND “resistance”[All Fields] AND “microbial”[All Fields]) OR “microbial drug resistance”[All Fields] OR (“drug”[All Fields] AND “resistance”[All Fields] AND “microbial”[All Fields]) OR ”drug resistance, microbial”[All Fields])	
Free text articles	Open access	

The terms of the MeSH hierarchy were adopted for antimicrobial resistance, considering that the terms MeSH is a controlled vocabulary of biomedical terms whose elements are assigned to a document by indexers (specialists in biomedical subjects) based on its context. They contain high-density document-wide information that cannot be deduced from the title or abstract that PubMed returns using keywords (NLM: Medical Subject Headings (MeSH)).

Then, a list of PMCIDs (unique identifiers provided by PubMed Central to each document) is generated to be used to access the full texts of articles through the E-Fetch utility.

Named entity recognition and knowledge discovery

The entity recognition and knowledge discovery process consist of two steps: in the first step (Fig. 1C), the Doc2Vec unsupervised learning algorithm from the Gensim library was used, which implements the Paragraph Vector–Distributed Memory model, to obtain the embedding of the retrieved documents (dense representation of a sequence of words). Table 2 displays the parameters used in the algorithm.

Table 2 Parameters Doc2Vec algorithm.

Parameters	Value	Description	
vector_size (int, optional)	300	Dimensionality of the feature vectors	
alpha (float, optional)	0.025	The initial learning rate	
min_alpha (float, optional)	0.00025	Learning rate will linearly drop to min_alpha as training progresses	
workers (int, optional)	18	Use these many worker threads to train the model (= faster training with multicore machines)	
min_count (int, optional)	3	Ignores all words with total frequency lower than this	
epochs (int, optional)	30	Number of iterations (epochs) over the corpus	
dm ({1,0}, optional)	1	Defines the training algorithm. If dm = 1, ‘distributed memory’ (PV-DM) is used. Otherwise, distributed bag of words (PV-DBOW) is employed	

In the second step (Fig. 1D), the pre-trained model was used, capable of predicting whether a set of documents {Doc1, Doc2, Doc3…Docn} belong to the context of a central document Doc0, to infer the similarity of the documents to the AMR context, represented by 4,290 terms extracted from CARD and the Gene Ontology Database (Ashburner et al., 2000), calculating the cosine distance (Nguyen & Bai, 2011) between them. The resulting value varies in the range between −1 and 1 where the higher the number, the greater the similarity with the context.

Finally, each of the scientific articles was automatically labeled (Fig. 1E), considering the arithmetic average of the cosine similarity, values above the average were defined as relevant and below the mean as irrelevant.

Evaluation of the proposed method

To evaluate the classification performance of the proposed TM approach (Fig. 2A), the same dataset was labeled using the Bag of words text representation model (Fig. 2B), adopting the specific vocabulary for AMR as a vector of features, and the documents were classified as relevant or irrelevant through the arithmetic mean of the weights of the words, obtained with the TF-IDF weighting method, similar to the methodology proposed in Suomela & Andrade (2005). Finally, the two (automatically) labeled databases were compared with a test database labeled by experts (Fig. 2C), who independently labeled the articles as relevant or irrelevant. Only the samples where the three experts converged on the labels were included in the test dataset.

Figure 2 Evaluation of the proposed method.

Predictions with automatically labeled data

To evaluate the efficiency of the proposed approach, which uses neural embeddings for labeling, the generated datasets (Fig. 3: Dataset_1 and Fig. 3: Dataset_2) were used as input to the supervised classifier SVM (Boser, Guyon & Vapnik, 1992; Drucker et al., 1997). The classification performance was evaluated through the analysis of the Precision, Recall, Accuracy, and F-Measure metrics (Grandini, Bagli & Visani, 2020), calculated from the test base labeled by experts and not used to train the SVM models (Fig. 3).

Figure 3 Performance of predictions with automatically labeled data.

For the SVM classifier, the feature/attribute vector (AMR vocabulary with 4,290 words) was weighted using the TF-IDF technique, and cross-validation, with 5-folds (default value of the adopted algorithm), was applied to explore the combination of parameters for determining the best model, as the effectiveness of the SVM depends on the kernel selection, which is the function that will be used by the algorithm, in the margin parameter (C), which determines a balance between maximizing the margin and minimizing classification errors, and the Gamma parameter, when the chosen kernel is Gaussian (or RBF) (Syarif, Prugel-Bennett & Wills, 2016), adopted in this experiment.

Data avaliability

The dataset, script, and models generated by this work are available at https://github.com/engbiopct/TextMiningAMR, under CC BY 4.0 Copyright license, with information regards the workflow adopted, a short step-by-step guide to the readers reproduce this experiment and the complementary materials.

Results and Discussion

Labeling

A collection of 88,300 scientific articles on antimicrobial resistance was retrieved from Pubmed Central, using the terms of reference of the MeSH (Medical Subject Headings, developed at the National Library of Medicine) hierarchical vocabulary referring to the AMR domain (https://meshb.nlm.nih.gov/record/ui?ui=D004352).

The retrieved dataset was submitted to the PV-DM text representation model, with the embedding of the documents obtained. The similarity of the documents to the AMR context (Table S1) was inferred, and the label “relevant” was automatically assigned (class 0) to all articles whose cosine distance value was equal to or greater than the arithmetic average of the cosine distances of the entire corpus, and the label “irrelevant” (class 1) to all articles with a cosine distance value lower than that referred to average, resulting in 43,136 records labeled relevant and 45,164 labeled irrelevant (Table S2).

The same initial dataset was submitted to the Bag of words text representation model in order to obtain the weights of the words in the documents according to the AMR dictionary and thus automatically assign the label “relevant” (class 0) to all articles with weight equal to or greater than the arithmetic average of the weights, and the label “irrelevant” (class 1) to all articles with a value lower than the mean, resulting in 45,946 records labeled as relevant and 42,354 labeled as irrelevant (Table S3).

With the two labeled datasets, the results were compared with a test dataset labeled by experts, consolidated with a total of 62 scientific articles, 15 labeled as relevant and 47 labeled as irrelevant (Table S4).

In the comparison, the proposed method labeled 44 articles according to the experts, which represents 71% of hits in general, with 80% of hits for the relevant label and 68% of hits for the irrelevant label. As for the baseline method, there were only 26 labels according to the experts, which represents 42% of overall performance, with 66% of correctness for the relevant label and 34% of correctness for the irrelevant label.

The proposed approach presents a superior performance about the baseline, which despite its simplicity, efficiency, and often surprising precision, does not take into account the order and semantics of the words, that is, the distances between them. This means that the words “mighty”, “strong” and “Paris” are equally distant. Although semantically, “powerful” is closer to “strong” than “Paris” (Le & Mikolov, 2014), characteristics present in the PV-DM model and fundamental human skills in manual data labeling tasks.

Classification

The two labeled datasets were submitted to the supervised SVM classifier, excluding the test dataset labeled by experts from training.

Figure 4 presents the confusion matrix of the SVM_1 classifier, trained with data from dataset 1 (Fig. 3: Dataset_1). There is a high degree of precision both in terms of true positives (relevant articles classified as relevant) and true negatives (non-relevant articles classified as non-relevant).

Figure 4 SVM classifier confusion matrix for dataset_1 (PV-DM).

Only one article was incorrectly classified as relevant (false positive), and five articles were incorrectly classified as irrelevant (false negative).

Figure 5 presents the confusion matrix of SVM_2, trained with data from dataset 2 (Fig. 3: Dataset_2), where a lower degree of precision is observed for both true positives and true negatives in relation to SVM_1. With this classifier, however, there was an increase in the number of incorrect classifications, with 33 articles incorrectly classified as relevant (false positives) and six articles incorrectly classified as irrelevant (false negatives).

Figure 5 SVM classifier confusion matrix for dataset_1 (Bag of Words).

Table 3 presents the results of the evaluation metrics: precision, recall, accuracy and the f-measure, obtained based on the results of the confusion matrix of the two classifiers. The results of the SVM_1 classifier were superior to the SVM_2 classifier in all evaluated metrics.

Table 3 Classifier performance assessment.

	Class	Precision	Recall	F1-score	Support	Accuracy	
SVM_1 (Doc2Vec+Mean)	0	0.74	0.93	0.82	15	0.90	
1	0.98	0.89	0.93	47	
SVM_2 (TF-IDF+Mean)	0	0.21	0.60	0.32	15	0.37	
1	0.70	0.30	0.42	47	

Accuracy, a metric that represents the overall performance of a model, reached 90% of accuracy and the f-measure, which is a harmonic average between the precision and recall metrics and that can be used as a single measure to represent the quality in the text mining (Rodriguez-Esteban, 2009) reached 82% success rate for the positive class and 93% for the negative class.

The results show that the best performance was obtained with the database labeled by the PV-DM model.

Table 4 presents the percentage of correct predictions, both in the labeling and in the classification stage, in comparison with the data labeled by experts and validates the hypothesis that the use of Paragraph Vector, Distributed Representations of Sentences and Documents associated with similarity with a specific context is able not only to perform the binary classification of large volumes of data but also to optimize the percentage of hits of supervised classifiers. The SVM_2 classifier showed a reduction in the number of hits compared to the Labeling step, although we adopted the same attribute vector and the same representation in both experiments (bag of words, weighted with TF-IDF).

Table 4 Results of labeling and classification steps vs experts.

	Relevant (%)	Irrelevant (%)	
Labeling			
Dataset_1	80	68	
Dataset_2	66	34	
Classification			
SVM_1	93	89	
SVM_2	60	29	

Conclusions

The proposed TM approach proved capable of identifying and prioritizing documents in the AMR context, as well as predicting the relevance of new documents in the same context. For this, we used the TM steps summarized in Fleuren & Alkema (2015), plus some specifics of the proposal such as (1) use of the MeSH hierarchy; (2) use of full text; (3) use of domain-specific dictionaries (CARD and Gene Ontology), fundamental in this process, as it facilitated the detection of similarity of articles to the AMR context and (4) adoption of unsupervised learning for better representation of texts. Additionally, we submitted the labeled bases to the SVM classifier to evaluate their performance in comparison to the test base labeled by human experts.

The proposed approach efficiently identifies scientific articles relevant to the AMR context. Therefore, it is a valuable tool to automate information capture processes from robust bibliographic reference databases such as Pubmed Central, as well as to accelerate the screening of documents with biocuration potential, facilitating the other stages of the biocuration process.

This work presents a new set of pre-trained document embeddings in the AMR domain and a base labeled for relevance according to similarity to CARD and Gene Ontology, which, in future work, can be used as input to other algorithms of supervised learning and for biocuration tools aimed at the identification and normalization of bioentities, detection of annotation events and filling out specific databases for AMR. This proposal is limited to the previous existence of an accurate database representing the main terms related to the target.

Supplemental Information

Supplemental Information 1 Terms from CARD, a rigorously curated collection of characterized, organized by the Antibiotic Resistance Ontology (ARO) and AMR gene detection models, and Gene Ontology databases, the knowledgebase world’s largest source of information on the functions o.

Click here for additional data file.

Supplemental Information 2 The dataset labeled by the proposed model, which the label “relevant” was automatically assigned to all articles whose cosine distance value was equal to or greater than the arithmetic average of the cosine distances of the entire corpus, and the label “i”.

Click here for additional data file.

Supplemental Information 3 The dataset labeled by the Bag of words text representation model, which assigns label “relevant” to all articles with weight equal to or greater than the arithmetic average of the weights, and the label “irrelevant” to all articles with a value lower th.

Click here for additional data file.

Supplemental Information 4 The test dataset labeled by experts, consolidated with a total of 62 scientific articles and the classification performed by the two tried methods.

The proposed method labeled 44 articles according to the experts, which represents 71% of hits in general, with 80% of hits for the relevant label and 68% of hits for the irrelevant label. As for the baseline method, there were only 26 labels according to the experts, which represents 42% of overall performance, with 66% of correctness for the relevant label and 34% of correctness for the irrelevant label.

Click here for additional data file.

Additional Information and Declarations

Competing Interests

Author Contributions

Data Availability

Rommel T. Ramos is an Academic Editor for PeerJ.

Kelle Fortunato Costa conceived and designed the experiments, performed the experiments, analyzed the data, prepared figures and/or tables, authored or reviewed drafts of the paper, and approved the final draft.

Fabrício Almeida Araújo analyzed the data, authored or reviewed drafts of the paper, and approved the final draft.

Jefferson Morais analyzed the data, authored or reviewed drafts of the paper, and approved the final draft.

Carlos Renato Lisboa Frances analyzed the data, authored or reviewed drafts of the paper, and approved the final draft.

Rommel T.J. Ramos conceived and designed the experiments, analyzed the data, authored or reviewed drafts of the paper, and approved the final draft.

The following information was supplied regarding data availability:

The data is available at GitHub: https://github.com/engbiopct/TextMiningAMR.

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
