# Peer review of "Text mining for identification of biological entities related to antibiotic resistant organisms"

_PeerJ, doi:10.7717/peerj.13351_

## Round 0.1 · original submission · Major Revisions

The reviewers' opinions were mixed. One reviewer noted fundamental limitations of the manuscript. Please review their comments and do your best to address them. I will need to send it out for a second round of review and go from there.

Reviewer 1 ·

Basic reporting

Thank your for a very well-written manuscript. The language is clear and the writing style is concise. A pleasure to read. A few comments

* Something is wrong with your PubMed IDs in Table 1 - None of the texts in the "text" column matches the PubMed IDs. E.g. your best scoring article PMID: "7232201" is a case report in French. Apart from this, I'm don't really find Table 1 particularly informative - perhaps include the entire list as a supplementary?

* You mention that you are looking for "metadata that enrich the characterization of target sequences", but I am not entirely sure what the criteria for "relevant" are. Could you please elaborate?

In terms of the annotations listed starting on line 70, many of these annotations are already integrated in PubTator Central. Have you tried this database for your search?

The paragraph starting on line 117 seems a bit redundant. Consider removing?

I guess the search term you list on line 139 ("antibiotcs resistance") contains a typo?

Experimental design

Methodologically, the method seems sound and is well described. You are not the first to embark on this endeavor, though.

* Did you test existing methods, E.g. BioReader, MScanner, MedlineRanker, and others? In order to justify publication of another tool for text classification, you should comment on how your method stands out from existing methods.

* There's a MeSH term for antimicrobial resistance (https://meshb.nlm.nih.gov/record/ui?ui=D004352). May I suggest checking how well this performs as a classifier? Could be a nice baseline for you to compare your method to.

* It is a bit unclear how you defined the two classes (relevant and irrelevant). It seems like you calculated the distance to your dictionary and set a more or less arbitrary threshold to define relevant or irrelevant, or did you manually curate the classes in the training set? If you assigned classes based on distance to the dictionary, isn't the whole classification step a bit redundant? Couldn't you just calculate the distance of any new article, and if it was above the threshold, you classify it as relevant? Seems like you are trying to predict a metric you just showed can be calculated.

Validity of the findings

Seems like your method performs quite well. As commented above, I need some more information about your relevant/irrelevant classification. Without this, it is hard to evaluate whether your method performs as anticipated.

Perhaps add a discussion of why your method fails for some cases.

Reviewer 2 ·

Basic reporting

There is significant lack of justification, limitations, and context for this work in the field. There is not a professional article structure, and the R&D is structure is very lacking, does not connect back to current literature, and is minimal.

Additionally, Indeed AMR is a huge problem across many domains of science and has become a One Health problem. Importantly, it is not just a biological problem but a multi-domain problem. The paper hones in only on the biological perspective for most of it – this is problematic in keeping the domains, language use, and interdisciplinary initiatives siloed. It is needed to add in more context and discussion relevant to this concept of One Health and how that can/will impact the model created and outcomes. Or, while this discussion and domain component is more important to recognize and discuss, the authors could further justify the biological focus for the scope of this project and provide limitations in the results and discussion for this approach.

Experimental design

There is no discussion on the size of the scientific articles extracted from pubmed. How was that collection determined? How were those key search terms determined? Does this mean that only open access articles were used for mining? This is not adequately explained or justified.

The first line of the R&D states “Using as search criteria the keywords “antibiotcs resistance”, we had access to the content of 4,517 scientific articles from PubMed”. There are two problems with this sentence – one I am hoping that it is just a mistype (that occurs several times) in the paper for antibiotics (not antibiotcs) and two when you use just the search term antibiotic resistance on pubmed, there are over 200K+ papers. 91K+ are full free text. How was 4,517 obtained? Was there more search terms added? Were the terms from Table 2 utilized as well or were they pulled as the CARD terms from the set of 4000? I viewed the searchtermarticle.py and the google colab is not accessible and pulls from a personal google sheet. Perhaps this will be fixed after publication but this does not clarify the above questions and further highlights that antibiotic resistance was the only search term used, and does not make any of this step reproducible. All of this needs to be justified and explained both for methodology used (and a section on limitations as based on this search and open access/not open access) needs to be added as well as making the code actually workable to reproduce the results in the paper.

The breakdown of relevant versus not relevant. There is some discussion of this in the methods, but there are a lot of limitations to discuss and also a lack of explanation of a use case of this kind of model on any AMR related phrase where this could be used. What about this utility for a biologist? Is it mostly just based on the amount and occurrence of CARD terms in the articles? Is that really a robust way to determine relevant or not relevant? How did the use of this dictonary, as stated in the conclusions, facilitate similarity in binning?

Validity of the findings

It is a huge jump, with no justification, to state this method could be used for other biological topics. There is no implication of that in this work, there is minimal justification of this working for AMR.

Additional comments

Figure 3 has no axis labels

All the tables in the table legend are called table 1

---

## Round 0.2 · Minor Revisions

Thank you for working to address the reviewers' concerns. I received a follow-up response from the second reviewer who was overall fine with your changes but requested additional changes. Please address those and resubmit.

Reviewer 2 ·

Basic reporting

The authors have addressed many of my comments from the previous review and I thank them for the improvements in the methodology and presentation. I have three aspects that could be easily addressed as a minor revision and complete the work as based on my previous review.

I can agree with the scope of the work as noted in the response to reviewer. I would still suggest this being clearer and directly stating in the Intro/R&D the scope of this work so it is properly contextualized by your audience.

It needs to be mentioned in the text that the resources were open to the public aka open source/open access articles so it’s clear to and correctly interpreted by the reader.

Limitations of this work should be clearly mentioned in the R&D/conclusions.

Experimental design

See above

Validity of the findings

See above

---

## Round 0.3 · Minor Revisions

Thank you for working to address the reviewer's comments. I read the revised parts of the manuscript. Although it helps, I found the revised sentence toward the end of the Introduction to be long and difficult to understand. There are some grammar problems with it as well. Please rework this part.

In the sentence added at the end of the Conclusions section, I am not sure what is meant by "to reach a better efficiency." This could probably be removed. If not, please make this more clear.

---

## Round 0.4 · accepted · Accept

Thank you for addressing the reviewers' concerns. Thank you also for making your data and code readily available for other researchers to use.